# An Alternative Statistical Model for Predicting Salinity Variations in Estuaries

**Ronghui Ye [1,2,3], Jun Kong [1,\*], Chengji Shen [1], Jinming Zhang [1] and Weisheng Zhang [4]**

[1] State Key Laboratory of Hydrology-Water Resources and Hydraulic Engineering, Hohai University, Nanjing 210098, China; ronghuiye@foxmail.com (R.Y.); c.shen@hhu.edu.cn (C.S.); zhangjinming01@hhu.edu.cn (J.Z.)

[2] Pearl River Hydraulic Research Institute, Pearl River Water Resources Commission, Guangzhou 510611, China

[3] Southern Marine Science and Engineering Guangdong Laboratory (Zhuhai), Zhuhai 519000, China

[4] Key Laboratory of Coastal Disaster and Defense, Ministry of Education, Hohai University, Nanjing 210098, China; weisheng_zhang@163.com

\* Correspondence: kongjun999@126.com

**Abstract:** Accurate salinity prediction can support the decision-making of water resources management to mitigate the threat of insufficient freshwater supply in densely populated estuaries. Statistical methods are low-cost and less time-consuming compared with numerical models and physical models for predicting estuarine salinity variations. This study proposes an alternative statistical model that can more accurately predict the salinity series in estuaries. The model incorporates an autoregressive model to characterize the memory effect of salinity and includes the changes in salinity driven by river discharge and tides. Furthermore, the Gamma distribution function was introduced to correct the hysteresis effects of river discharge, tides and salinity. Based on fixed corrections of long-term effects, dynamic corrections of short-term effects were added to weaken the hysteresis effects. Real-world model application to the Pearl River Estuary obtained satisfactory agreement between predicted and measured salinity peaks, indicating the accuracy of salinity forecasting. Cross-validation and weekly salinity prediction under small, medium and large river discharges were also conducted to further test the reliability of the model. The statistical model provides a good reference for predicting salinity variations in estuaries.

**Keywords:** groundwater monitoring; saltwater intrusion; autoregressive model; Gamma distribution function; Pearl River Estuary

## 1. Introduction

Saltwater intrusion (SI) has posed a great threat to the freshwater supply for domestic, agricultural and industrial demands in densely populated estuaries. Better understanding of SI in estuaries is fundamental for properly managing water resources and ensuring freshwater supply [1].

SI involves complex saltwater–freshwater mixing, and the process is influenced by various factors, mainly including river discharge, tides, wind and sea-level rise (SLR). Many studies have confirmed a negative correlation between river discharge and the intensity of SI [2–4]. The salt transport associated with SI under different tidal mixing schemes may differ greatly in terms of timing [1]. Specifically, saltwater mainly intrudes landward during spring tides in well-mixed estuaries [5–7] and during neap tides in partially mixed estuaries [4,6,8]. The wind is also an important factor affecting SI, controlling estuarine circulation and stratification [9–12]. Traditionally, the wind is thought to promote vertical mixing [13,14]. Based on field observations at the York River, [15] discovered that the upstream wind can weaken salinity stratification, while the downstream wind might strengthen (lowintensity)

or weaken (highintensity) such stratification. SLR arising from global warming further aggravates the intensity of SI [16]. The nonlinear interactions between the above-mentioned factors make it difficult to accurately predict SI, thereby impeding the development and implementation of effective measures for water resources management.

To solve the problem above, researchers have proposed different methods to investigate and forecast SI in estuaries, such as laboratory experiments, analytical models and numerical simulations. Physical flume experiments were conducted to analyze the responses of turbulent mixing intensity and spatial salinity distribution to different topographies [17] and changes in SI length driven by different tidal ranges [18]. Results showed that there is a critical tidal range corresponding to the minimum SI length. If the tidal range is smaller than the critical value, the increase in tidal range results in a rapid decrease in SI length, while SI length slowly increases with the increase in tidal range if it is above the critical value [1]. One-dimensional (1D) analytical models have been derived to predict SI length [19–21], and various numerical simulations have been carried out to analyze SI intensity using estuarine oceanic hydrodynamic models based on structured or unstructured grids [22–24]. A high-precision TVD2 scheme was proposed to solve the advection term for the material transport equation, thereby improving the accuracy of salinity transport simulation [25]. The accuracy of salinity prediction using machine learning methods can be improved by optimizing various neural network methods [26–28]. Based on the relationship between the material and the salinity in the water, the remote sensing inversion method was established to forecast salinity [7,29]. Twentyyears of data collected in the Caloosahatchee River Estuary were used to calibrate and validate a statistical regression model [30]. A statistical model was developed to predict the salinity in the upper South Branch of the Yangtze River Estuary [31]. This model only requires two variables: runoff at Datong station and lunar calendar date.

Although the studies mentioned above promoted accurate salinity forecasting in estuaries, those research methods are costly and time-consuming. More importantly, existing studies have mainly focused on the short-term prediction of salinity and SI length [20–22,27], whereas studies targeting mid- to long-term prediction are rare [26,30,31]. Furthermore, the memory effects of river discharge, tidal range and salinity have also been ignored. Numerical models are among the most widely used methods, but the construction of models for estuaries is complex. The influence of boundary factors needs to be included in the simulation of hydrodynamics and salinity. However, changes to estuarine topography caused by human activities make it difficult for the model to satisfy the accuracy criterion of SI simulation. In comparison, statistical methods generalizing the characteristics of geomorphological evolution and mutual nonlinear effects among multiple dynamic factors have lower costs and are less time-consuming. Statistical methods can also realize the prediction of salinity changes in specific locations. A mid- to long-term salinity forecasting model can be constructed by analyzing the relationship between main driving forces (river discharge and tidal range) and SI intensity.

## 2. Study Area and Data Source

### 2.1. Study Area

Pearl River Estuary (PRE) (Figure 1), as the third largest river in China, is one of the most densely populated estuaries in the world. The river networks of the PRE cover approximately 9750 km$^2$, with a total length of more than 1600 km and a coastline extending over 450 km from east to west [32]. There are eight outlets of the Pearl River delta. The North River and the East River mainly flow into the Lingding Bay through four east outlets (Humen, Jiaomen, Hongqimen and Hengmen), while the West River mainly runs into the South China Sea and Huangmao Bay through four west outlets (Modaomen, Jitimen, Hutiaomen and Yamen). Such vast river networks lead to rather complicated hydrodynamics and salinity variations of the PRE.

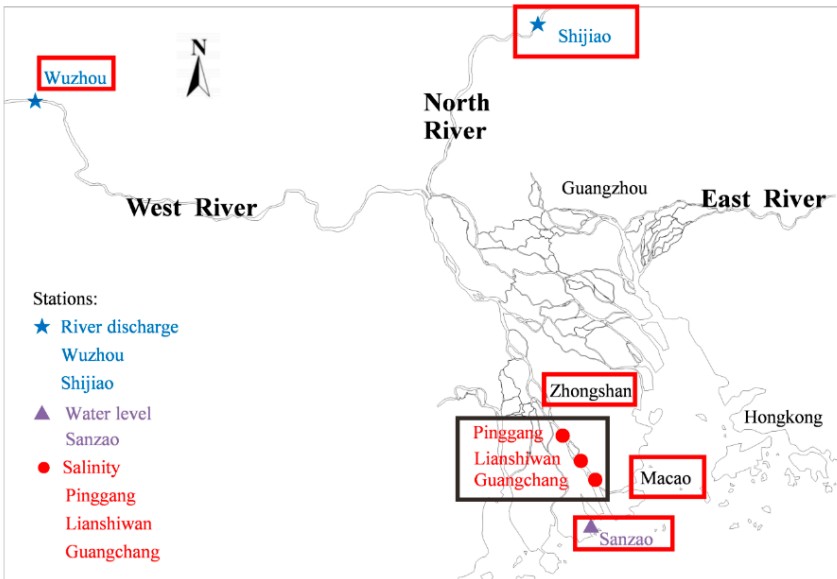

**Figure 1.** Location and geological map of the study area showing the locations of sampling sites.

Modaomen Channel, located in the lower reaches of the West River, was selected as the study area. The annual runoff of Modaomen Channel accounts for 26.6% of the total runoff of the PRE, making it the most important discharge channel. Meanwhile, the runoff distribution of Modaomen Channel is quite uneven in a year, with 75.7% of therunoff occurring during the flood season (from April to September) and 24.3% of the runoff occurringduring the dry season (from October to March). The tidal signal of Modaomen Estuary is micro (mean annual tidal range of 0.85 m) and irregular semidiurnal (average flood and ebb tidal durations are 5.37 h and 7.25 h, respectively) [Ye et al., 2017]. During the flood season, the intense runoff inhibits SI, while during the dry season, the reduced runoff allows seawater from the outer sea to move upstream, thereby intensifying SI. There are many water intakes along the Modaomen Channel (e.g., Zhuzhoutou pumping station, Pinggang pumping station and Guangchang pumping station), making it the main source of drinking water for Zhongshan, Zhuhai and Macao. However, in recent years, the reclamation projects at the entrance and the sand extraction in the upstream region have led toa deepening of the riverbed. As a result, the tides have been enhanced and the SI has been intensified, greatly threatening the upstream freshwater supply andcausing water from these intakes not to meet the standard of drinking water (the standard in China defined the salinity of drinking water should bebelow0.5). At thePinggang pumping station, as an example, the total duration of substandard water quality was 1612 h during 2005–2006, but this increased to 2117 hin 2011–2012.

*2.2. Data Source*

This study considered runoff and tide, the relative strength between which affects the intensity of SI. The river discharge was measured at Wuzhou station and Shijiao station, while water level data were collected at Sanzao station (Figure 1). Surface salinity at three stations was selected for validating the statistical model proposed in this study (model development provided in the next section), including Pinggang pumping station, Lianshiwan station and Guangchang pumping station. This paper focused on dry seasons, during which SI is more intense and salinity change is more variable, thereby better testing the ability of the model to accurately predict salinity variations. All the data were measured hourly from 2007 to 2008 and from 2011 to 2013 during dry seasons. Daily average river discharge/salinity data were obtained by averaging the hourly river discharge/salinity data, and the daily maximum tidal range was derived from hourly water level data. Unless specified, the river discharge, tidal range and salinity hereinafter refer to the processed values. These time-series data provide a valuable resource to validate the statistical model proposed for salinity prediction.

## 3. Model Development

### 3.1. Relationship between Salinity and River Discharge

Among all the factors controlling SI, runoff is a critical one. The comparison in Figure 2a shows that, despite the similar variation pattern, river discharge at Wuzhou station is generally larger than that at Shijiao station. From the comparison of time-series salinity in Pinggang, Lianshiwan and Guangchang pumping stations (Figure 2b), it is clear that salinity gradually increases from the upper to the lower reaches of the Pearl River.

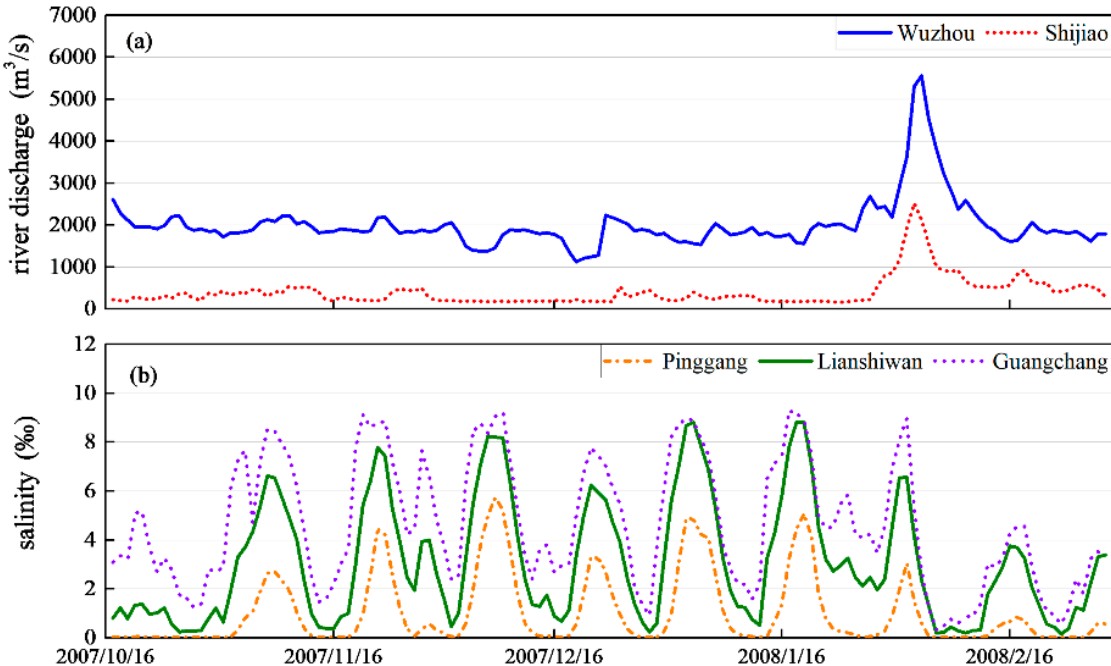

**Figure 2.** Dataof (**a**) daily average river discharge measured at Wuzhou station and Shijiao station and (**b**) daily average salinity measured at Pinggang, Lianshiwan and Guangchang pumping stations collected from October 2007 to February 2008.

To characterize the memory effects of river discharge on salinity, we introduced the Gamma distribution function, which has been widely used in hydrology [33–35] and is given as follows:

$$\omega_{t-i} = \text{Gamma}(\alpha, \beta, i) = \beta^\alpha \frac{1}{\Gamma(\alpha)} (i)^{\alpha-1} \exp(-\beta i) \tag{1}$$

where $t$ is the present day, $t - i$ is the $i$ day(s) before the present day, $\omega_{t-i}$ is the gamma coefficient of the previous day $t - i$, $\alpha$ is the shape factor and $\beta$ is the scale factor. The daily average river discharge of the previous 3 days at Wuzhou and Shijiao stations can be weighted according to Equation (2) below. To simplify the analysis, we put together the weighed river discharges at the two stations:

$$\widetilde{Q_t} = \frac{\sum_{i=1}^{3} \omega_{t-i} Q_{t-i}}{\sum_{i=1}^{3} \omega_{t-i}} \tag{2}$$

where $Q_{t-i}$ is the daily average river discharge of the previous day $t - i$ and $\widetilde{Q_t}$ is the weighed daily average river discharge.

The exponential decay function has been widely applied to characterize the relationship between river discharge and salinity [30,31,36]. However, regression analysis in Figure 3 shows small coefficients of determination ($R^2$) for the three stations, indicating that it is insufficient to simply consider the memory

effects of river discharge on salinity. Moreover, when the station is closer to the estuary, the effect of river discharge becomes weaker, as manifested by $R^2$ being reduced from 0.3586 to 0.2684 (Figure 3).

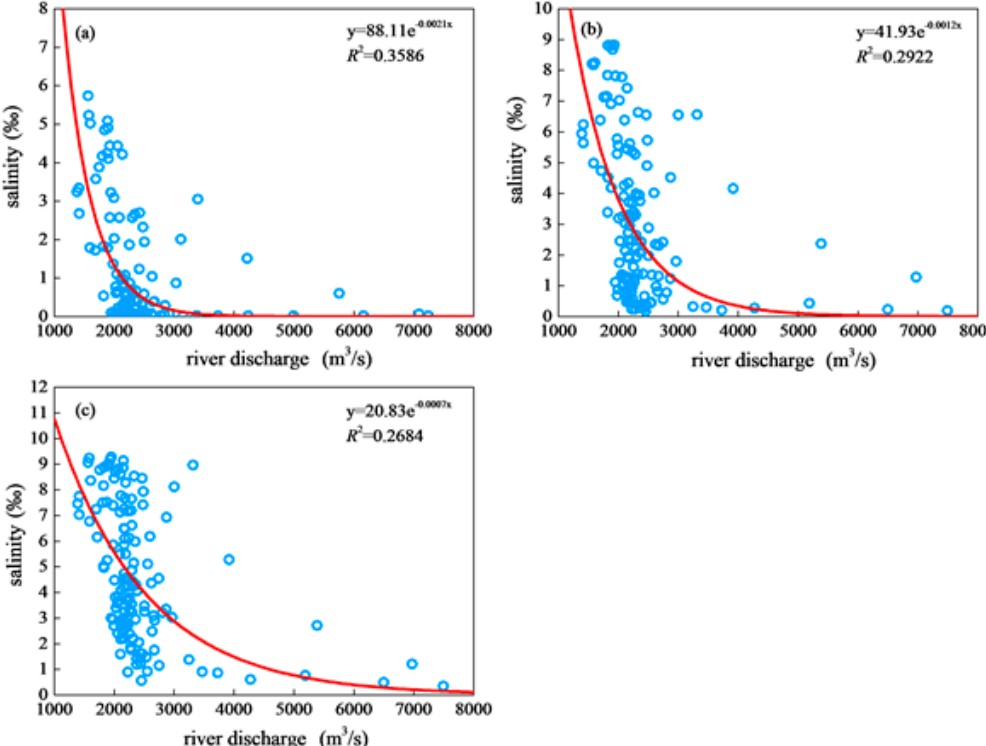

**Figure 3.** Regression between weighed daily average river discharge and daily average salinity at (**a**) Pinggang, (**b**) Lianshiwan and (**c**) Guangchang. Lines are the best-fit regression lines.

*3.2. Relationship between Salinity and Tidal Range*

Apart from river discharge, tidal range also greatly affects salinity distribution in estuaries [31,37]. As mentioned above, saltwater intrudes landward mainly during spring tides for well-mixed estuaries and during neap tides for partially mixed estuaries. Considering that the PRE is partially mixed, SI occurs primarily during neap tides. The field data of vertical salinity and velocity in the Modaomen Channel show that tidal current during neap tides is the main driving force of SI, which is the most intense during the transition periods from neap to spring tides [38,39]. Figure 4 shows a significant phase difference between the peaks of salinity and tidal range [40]. Since statistical methods cannot effectively characterize the relationship between tidal range and salinity, we integrated salinity changes driven by tidal range into the memory effect of salinity.

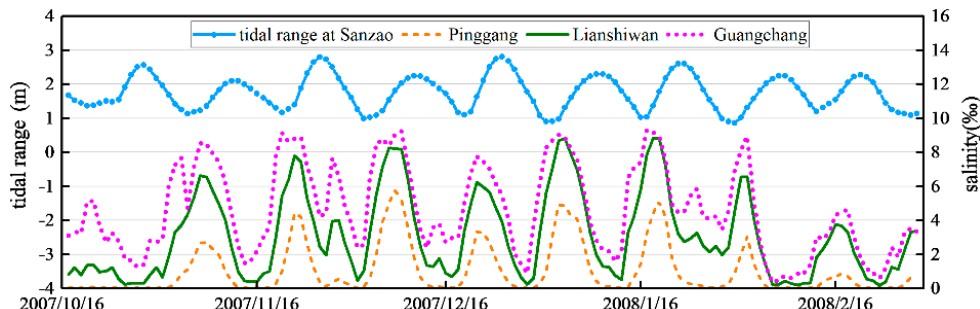

**Figure 4.** Time series of daily maximum tidal range measured at Sanzao and daily average salinity measured at Pinggang, Lianshiwan and Guangchang from October 2007 to February 2008.



### 3.3. Model Formulation

This study introduced an autoregressive model to account for the hysteresis effect of salinity:

$$S_t = F(S_{t-1}, S_{t-2}, S_{t-3} \cdots) \tag{3}$$

where $S_t$ is the salinity on day $t$ and $S_{t-i}$ is the salinity of the previous day $t - i$.

We also used the Gamma distribution function to analyze the long-term memory effects of salinity, based on the following corrections:

$$
\begin{aligned}
\xi_a &= \frac{\omega_{t-1}}{\sum_{i=1}^{3} \omega_{t-i}} \\
\xi_b &= \frac{\omega_{t-2}}{\sum_{i=1}^{3} \omega_{t-i}} \\
\xi_c &= \frac{\omega_{t-3}}{\sum_{i=1}^{3} \omega_{t-i}}
\end{aligned}
\tag{4}
$$

where $\xi_a$, $\xi_b$ and $\xi_c$ are the correction coefficients of memory effect based on long-time series of salinity, which reflect the effects of long-term changes in SLR and estuarine topography. The values of $\alpha$ and $\beta$ for calculating the Gamma coefficients are 1 and 8.6 for Pinggang pumping station, 1 and 6.2 for Lianshiwan station and 1 and 4.8 for Guangchang pumping station. With the correction coefficients determined, the daily average salinity of the previous 3 days at the three stations can be corrected according to the following formula:

$$\widetilde{S_t} = \xi_a S_{t-1} + \xi_b S_{t-2} + \xi_c S_{t-3} \tag{5}$$

where $\widetilde{S_t}$ is the predicted salinity on day $t$.

Despite the correction, the predicted salinity is still inaccurate and lags behind the measured daily average salinity (Figure 5). To further improve accuracy, we incorporated the dynamic corrections of short-term effects into the fixed corrections of long-term effects:

$$
\begin{aligned}
\widetilde{S_t} &= \xi_a S_{t-1} + \xi_b S_{t-2} + \xi_c S_{t-3} \\
S_{t-2} &= \phi_1 \xi_a S_{t-3} + \phi_2 \xi_b S_{t-4} + \phi_3 \xi_c S_{t-5} \\
S_{t-3} &= \phi_1 \xi_a S_{t-5} + \phi_2 \xi_b S_{t-6} + \phi_3 \xi_c S_{t-7}
\end{aligned}
\tag{6}
$$

where $\phi_1$, $\phi_2$ and $\phi_3$ are the correction coefficients of memory effect based on short-time series of salinity, which explain the effects of short-term changes in tidal range, wind and water diversion in the upper reaches and other anthropogenic activities. Based on Equation (6), $\widetilde{S_t}$ in Equation (5) can be rewritten as

$$S_{t-3} = \phi_1 \xi_a S_{t-5} + \phi_2 \xi_b S_{t-6} + \phi_3 \xi_c S_{t-7} \tag{7}$$

The comparison in Figure 5 confirms the improved prediction accuracy based on Equation (7). However, the predicted salinities are higher than the measurements. Given the poor accuracy of salinity prediction using either river discharge or memory salinity alone, we developed a statistical model that considers the memory effects of both factors:

$$S_{t-3} = \phi_1 \xi_a S_{t-5} + \phi_2 \xi_b S_{t-6} + \phi_3 \xi_c S_{t-7} \tag{8}$$

where $S_t$ is the predicted salinity on day $t$ and *A*, *B* and *C* are the correction coefficients.

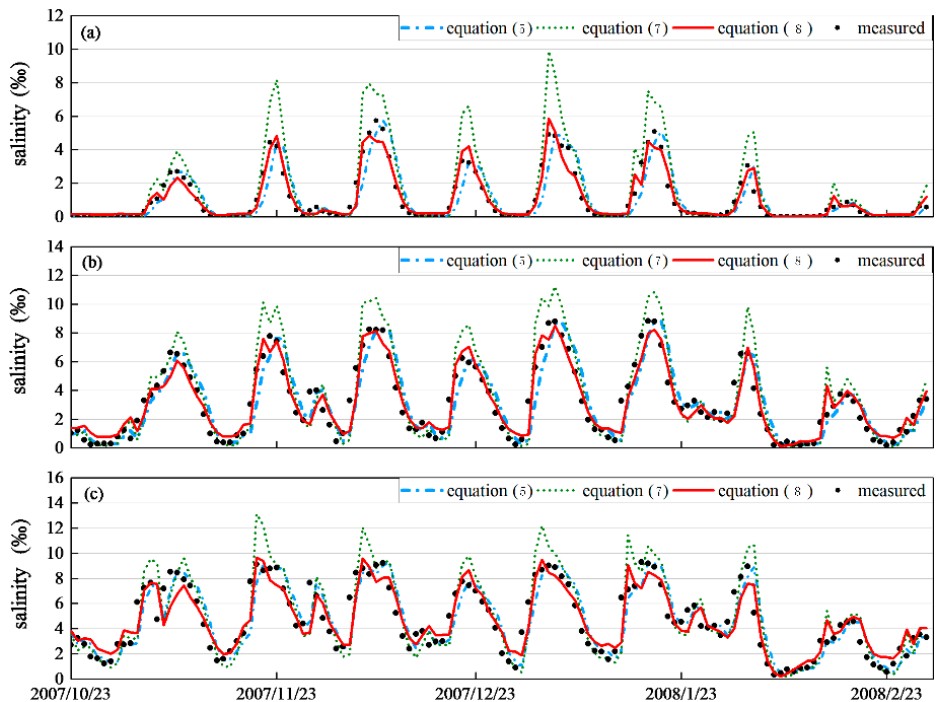

**Figure 5.** Comparison of salinity calculated by Equations (5), (7) and (8) and measured salinity at
(**a**) Pinggang, (**b**) Lianshiwan and (**c**) Guangchang.

## 4. Model Calibration, Application and Further Tests

### 4.1. Model Calibration

To obtain the values of correction coefficients *A*, *B* and *C* in Equation (8), we calibrated the
statistical model using salinity data collected during October 2007 and February 2008 (values listed in
Table 1). The comparisons in Figures 5 and 6 show slight differences between the measured salinity
variations and predicted values based on Equation (8), indicating that considering the memory salinity
can improve prediction accuracy. It should be noted that the measured maximum salinity values are in
good agreement with the predicted ones due to the correction of the long- and short-term effects that
weakened the hysteresis effects.

**Table 1.** Coefficients of different locations for model calibration.

| Location | *A* | *B* | *C* |
|----------|-----|-----|-----|
| Pinggang | 2.4429 | −0.0013 | 0.5711 |
| Lianshiwan | 3.1579 | −0.0006 | 0.6701 |
| Guangchang | 4.8074 | −0.0005 | 0.6349 |

To further evaluate the performance of the model, we used three statistical parameters: (1) the
coefficient of determination ($R^2$), an indicator of the percent of variation of the measured salinity
explained by the predicted salinity; (2) the root-mean-square error (*RMSE*), a measure of the deviation of
the predicted salinity from the measured salinity; and (3) the Nash–Sutcliffe efficiency coefficient (*NSE*),
which indicates how well the plot of observed versus predicted data fits the 1:1 linear regression line.

The *NSE* may range between $-\infty$ and 1, with 1 being the optimal value [30]. Mathematical equations of $R^2$, *RMSE* and *NSE* are givenas follows:

$$R^2 = \left[ \frac{\sum_{i=1}^n (M_i - \overline{M})(P_i - \overline{P})}{\sqrt{\sum_{i=1}^n (M_i - \overline{M})^2} \sqrt{\sum_{i=1}^n (P_i - \overline{P})^2}} \right]^2 \tag{9}$$

$$RMSE = \frac{\sqrt{\sum_{i=1}^n (M_i - P_i)^2}}{n} \tag{10}$$

$$NSE = 1 - \frac{\sum_{i=1}^n (M_i - P_i)^2}{\sum_{i=1}^n (M_i - \overline{M})^2} \tag{11}$$

where $n$ is the number of data, $M_i$ is the measured salinity, $\overline{M}$ is the mean of the measured salinity, $P_i$ is the predicted salinity and $\overline{P}$ is the mean of the predicted salinity.

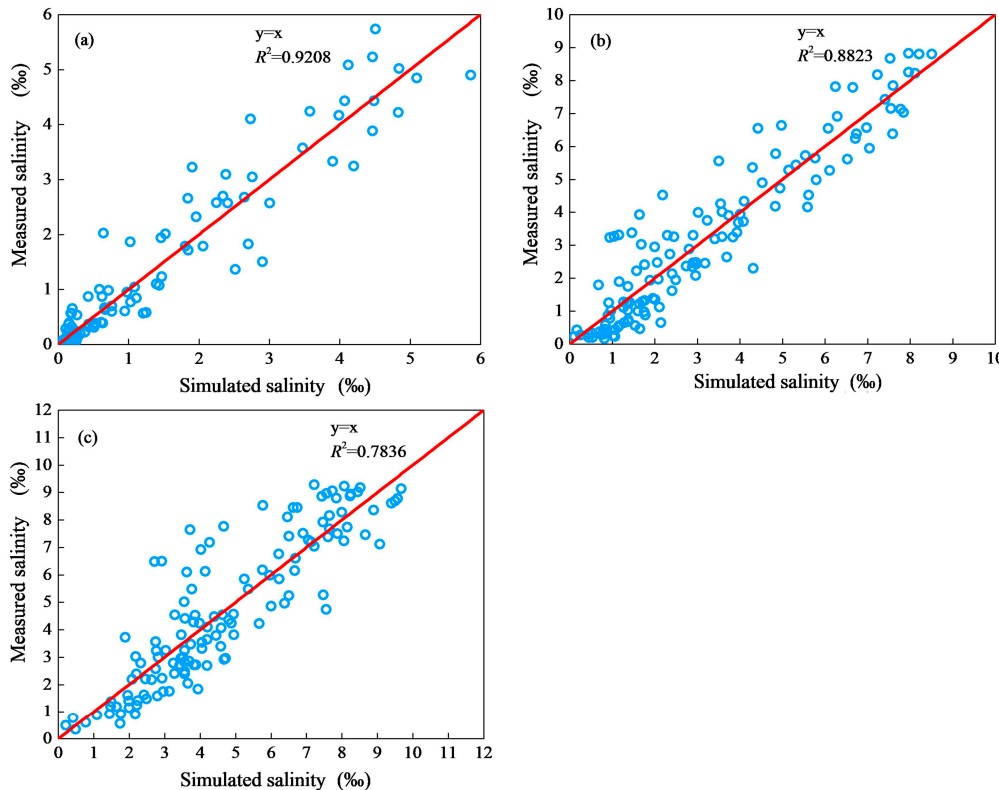

**Figure 6.** Plots of measured salinity versus simulated salinity at (**a**) Pinggang, (**b**) Lianshiwan (**c**) Guangchang during the model calibration period. Lines are the best fit 1:1 regression lines.

Table 2 lists the statistical parameters of model calibration for Pinggang, Lianshiwan and Guangchang pumping stations. Overall, the values of $R^2$, *RMSE* and *NSE* show the improved prediction accuracy of the model. For example, $R^2$ gradually increases to 0.9208 from downstream to upstream, and the *RMSE* ranges from 0.4201 at Pinggang pumping station to 1.2363 at Guangchang pumping station. Tidal dynamics mainly affect the salinity of the upstream river and barely exert influence on that of the downstream river. Therefore, the hysteresis effect on the salinity of the upstream river is stronger than that on the salinity of the downstream river.

**Table 2.** Summary statistics of model calibration from 2007 to 2008 and model application from 2011 to 2013.

| Location | $n$ | $\bar{O}$ (‰) | $O_{min}$ (‰) | $O_{max}$ (‰) | $R^2$ | RMSE (‰) | NSE |
|---|---|---|---|---|---|---|---|
| Pinggang | | | | | | | |
| Calibration (2007–2008) | 130 | 1.0847 | 0.0146 | 5.7379 | 0.9208 | 0.4201 | 0.9207 |
| Application (2011–2012) | 176 | 1.1040 | 0.0212 | 5.7767 | 0.9195 | 0.5066 | 0.8750 |
| Application (2012–2013) | 154 | 0.1622 | 0.0158 | 2.6646 | 0.8560 | 0.1551 | 0.8377 |
| Lianshiwan | | | | | | | |
| Calibration (2007–2008) | 130 | 3.1935 | 0.1633 | 8.8179 | 0.8823 | 0.8553 | 0.8822 |
| Application (2011–2012) | 142 | 2.8297 | 0.0053 | 9.2096 | 0.8904 | 0.8946 | 0.8882 |
| Guangchang | | | | | | | |
| Calibration (2007–2008) | 130 | 4.7043 | 0.3429 | 9.2829 | 0.7836 | 1.2363 | 0.7876 |
| Application (2011–2012) | 176 | 5.2052 | 0.0173 | 15.9654 | 0.8617 | 1.4963 | 0.8503 |

*4.2. Model Application*

We applied the calibrated model to predict the salinity measured from September 2011 to February 2012 and from September 2012 to February 2013 at Pinggang, Lianshiwan and Guangchang pumping stations. Table 2 lists the values of $R^2$, *RMSE* and *NSE*. For the three stations, the values of $R^2$ all reach 0.85, indicating a good match between the measured and predicted salinities. The agreement is further demonstrated by the comparison of salinity measurements and predictions in Figure 7. These results of real-world model application prove that the statistical model considering the memory effects of river discharge and salinity is more accurate.

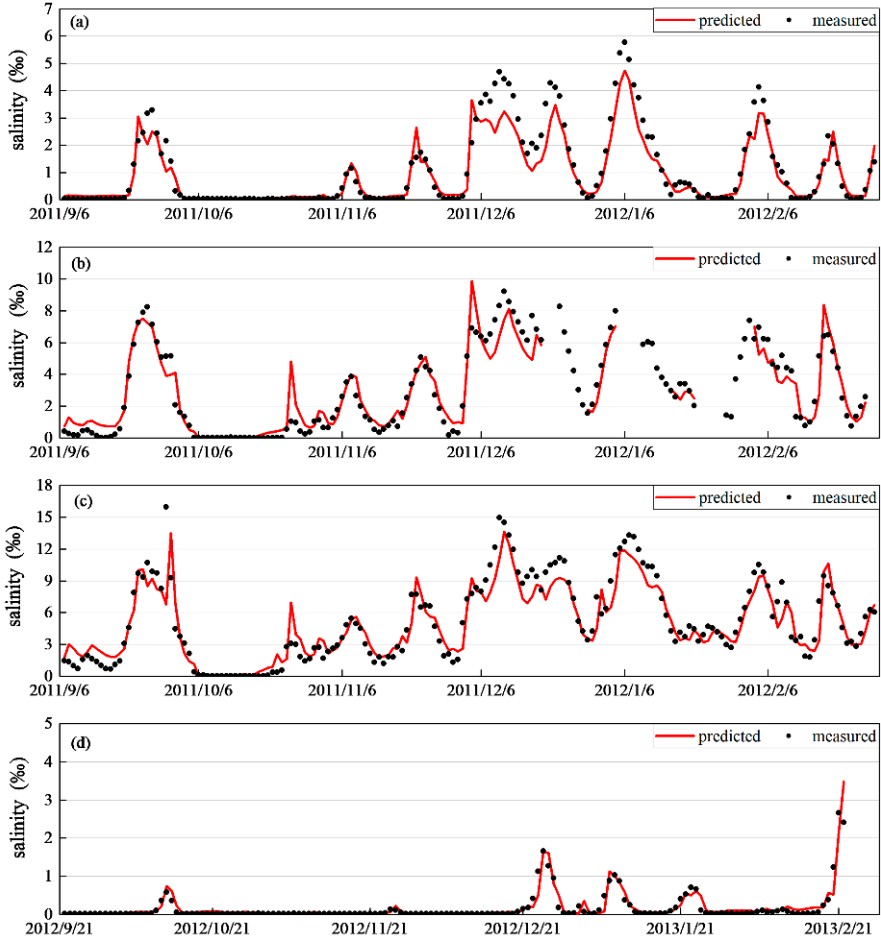

**Figure 7.** Comparison of measured and predicted salinity at (**a**) Pinggang, (**b**) Lianshiwan, (**c**) Guangchang and (**d**) Pinggangfrom September 2012 to February 2013.

### 4.3. Further Tests

#### 4.3.1. Sensitivity Analysis of River Discharge

As a dominant factor affecting SI, runoff can vary greatly under anthropogenic influence. Based on the Guangchang pumping station, we conducted a sensitivity analysis by increasing and decreasing the river discharge by 50%, with the other factors remaining unchanged. The model developed in the previous section was used:

$$S_Q = A exp\left(B\widetilde{Q_t}\right) \tag{12}$$

where $S_Q$ is the predicted salinity contributed by river discharge.

Figure 8 shows that when river discharge increases by 50%, $S_Q$ changes considerably, whereas $S_t$ varies slightly. In contrast, with a 50% reduction in river discharge, both $S_Q$ and $S_t$ increase notably. The comparison indicates that, when river discharge is small, salinity variation at estuaries is more sensitive to the changes. While under conditions of large river discharge, salinity variation becomes less sensitive.

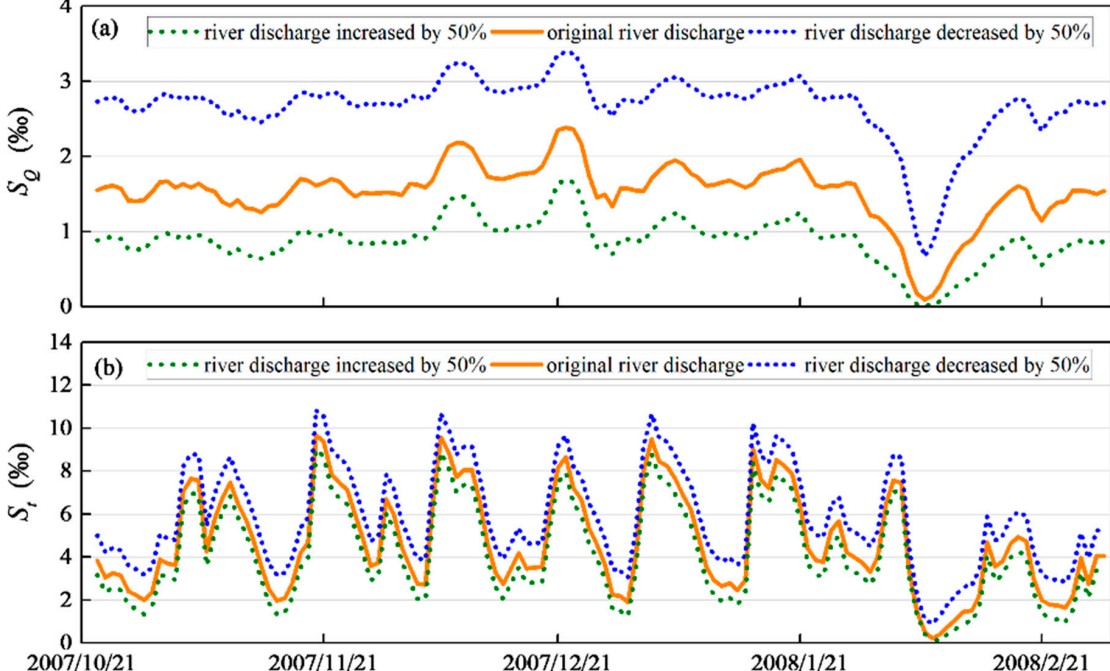

**Figure 8.** Comparison of $S_Q$ and $S_t$ of different river discharges at Guangchang station. (**a**) $S_Q$ of different river discharges at Guangchang station (**b**) $S_t$ of different river discharges at Guangchang station.

#### 4.3.2. Cross-Validation

Salinity data collected from Pinggang, Lianshiwan and Guangchang stations were used for cross-validation. Data from September 2011 to February 2012 were used for model calibration, while data from October 2007 to February 2008 and from September 2012 to February 2013 were used for model application. Due to the lack of salinity data at Lianshiwan station from 2011 to 2012, the prediction was conducted based on data at Pinggang and Guangchang pumping stations. The comparison between measured and predicted salinities (Figures 9 and 10) and statistical parameters ($R^2$, *RMSE* and *NSE*) of model calibration and application (Table 2) show that the prediction results are satisfactory, further demonstrating the improved model performance.

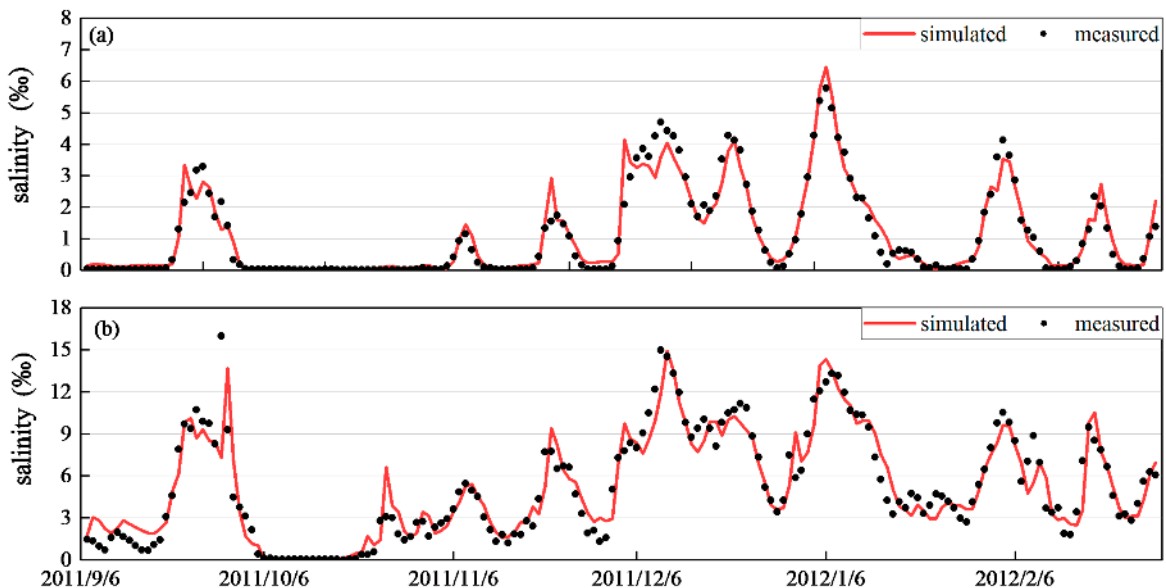

**Figure 9.** Comparison of measured and predicted salinity at (**a**) Pinggang station and (**b**) Guangchangstation from September 2011 to February 2012.

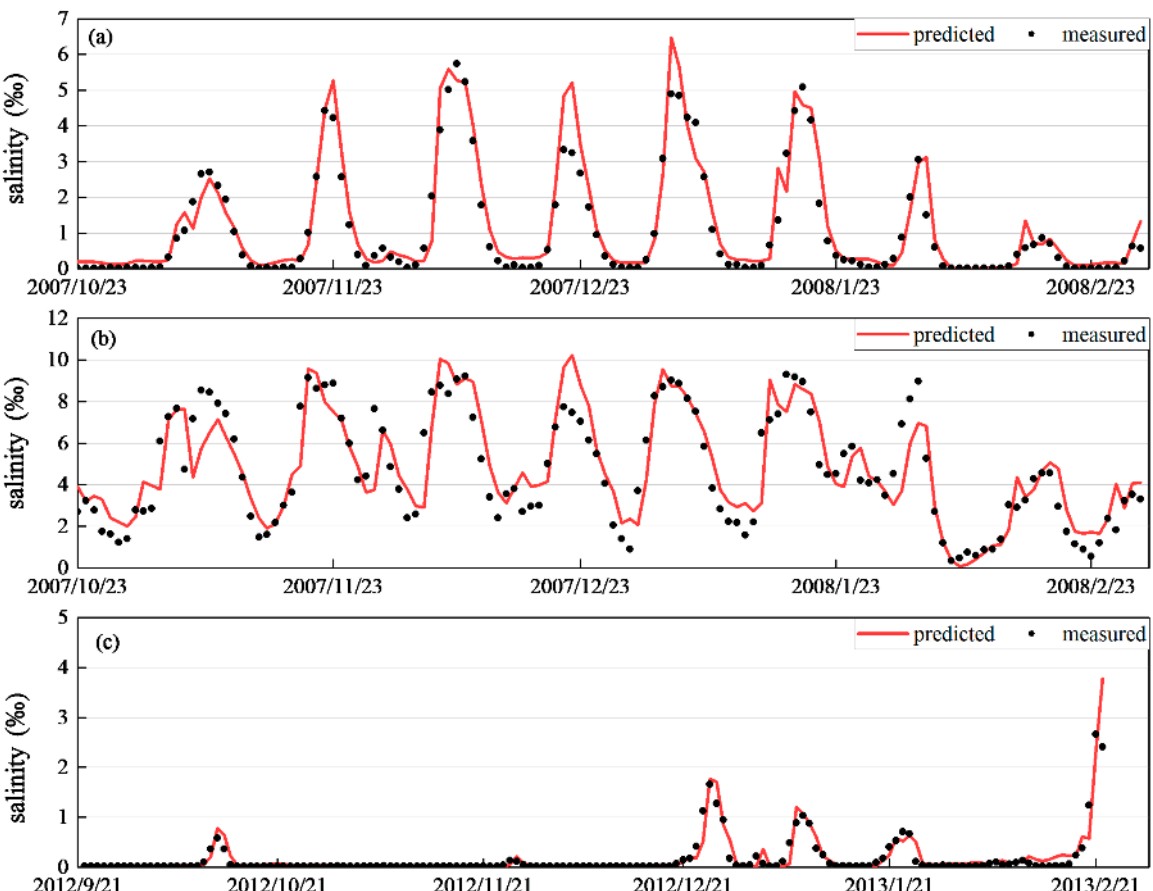

**Figure 10.** Comparison of measured and predicted salinity at (**a**) Pinggang, (**b**) Guangchangand (**c**) Pinggang stations.

### 4.3.3. Analysis of Weekly Prediction

A representative analysis was conducted at the Guangchang pumping station. The reliability of predicted salinity for the next 7 days was studied under three river discharges: 1400 m³/s (small), 2000 m³/s (medium) and 9000 m³/s (large). Figure 11 shows that salinity prediction for the next 7days in all three cases is relatively accurate. In particular, when river discharge is large enough (e.g., 9000 m³/s in Figure 11c), it almost completely prevents SI, and the salinity for the next 7 days is close to 0‰, which can be replicated by the statistical model.

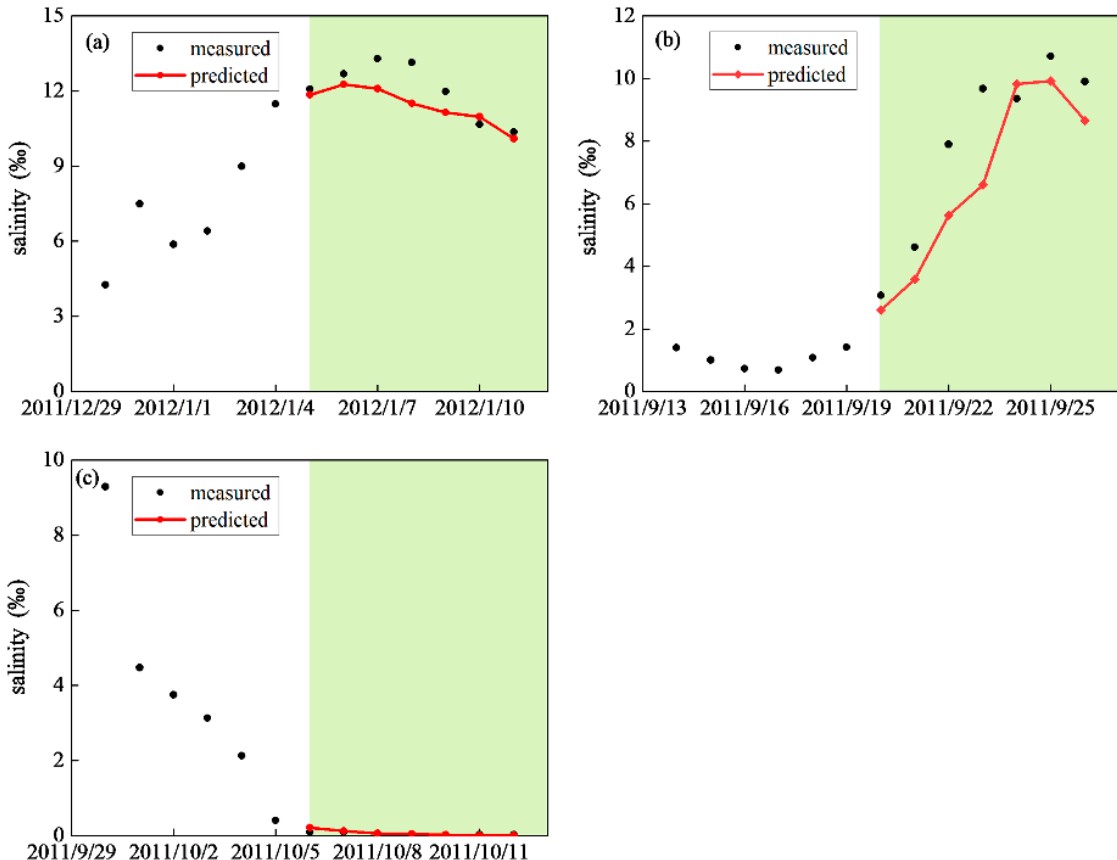

**Figure 11.** Comparison of measured and predicted salinity of the upcoming 7 days under conditions of (**a**) small, (**b**) medium and(**c**) large river discharge at Guangchang station.

### 4.3.4. Analysis of Short-Term Time-Series

The prediction model in some estuaries where measured data are inadequate can only be constructed with limited short-term data. In this section, data from October 2007 to December 2007 were used for model calibration while data from January 2008 to February 2008 were used for model application. The comparison between measured and predicted salinities presented in Figures 12 and 13 and statistical parameters listed in Table 3 show satisfactory prediction. More importantly, the results show that the statistical model proposed in this study can be applied to predicting salinity variations at estuaries with not only long-term data but also short-term data.

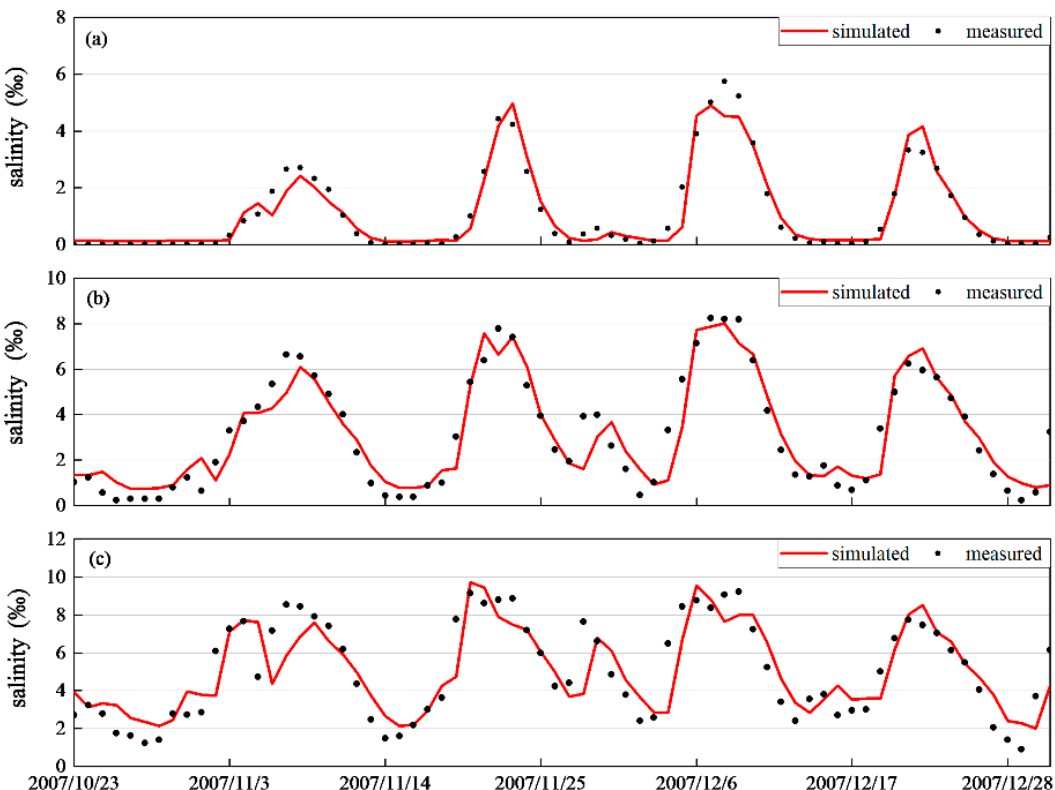

**Figure 12.** Comparison of measured and simulated salinity at (**a**) Pinggang, (**b**) Guangchang and (**c**) Pinggang stationsfrom October 2007 to December 2007.

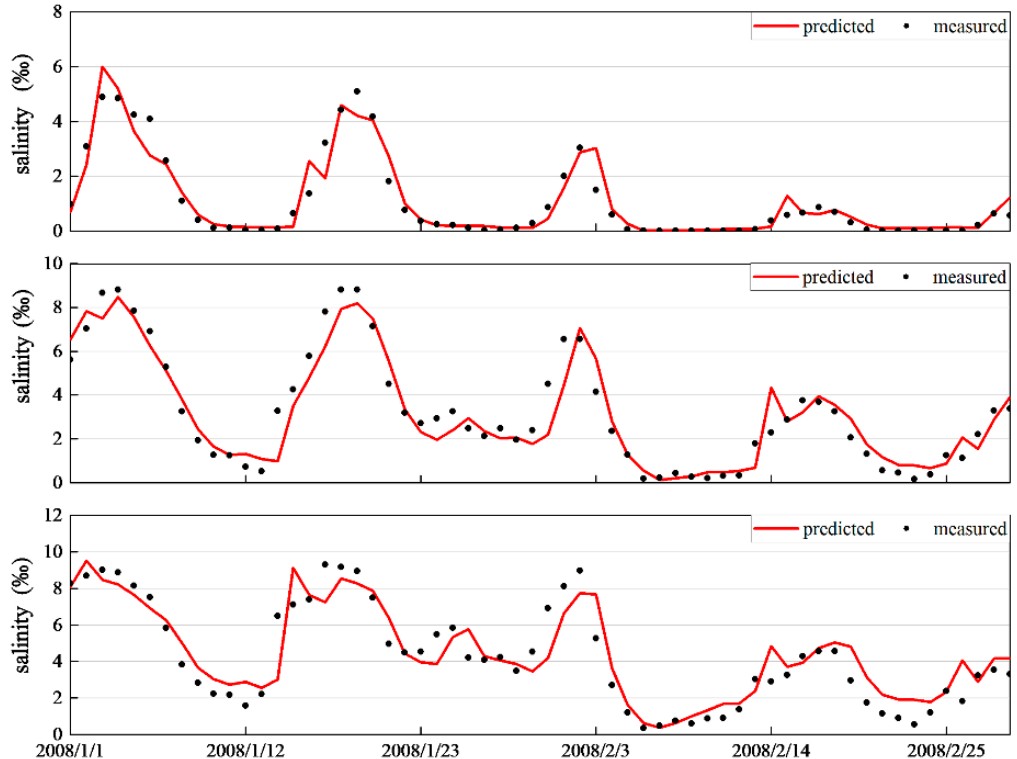

**Figure 13.** Comparison of measured and predicted salinity at (**a**) Pinggang, (**b**) Guangchang and (**c**) Pinggang stations from January 2008 to February 2008.

**Table 3.** Summary statistics of model performance for model calibration from October 2007 to December 2007 and model application from January 2008 to February 2008.

| Location | $n$ | $\bar{O}$ (‰) | $O_{min}$ (‰) | $O_{max}$ (‰) | $R^2$ | *RMSE* (‰) | *NSE* |
|---|---|---|---|---|---|---|---|
| Pinggang | | | | | | | |
| Calibration (23 October 2007 to 31 December 2007) | 70 | 1.1142 | 0.0200 | 5.7379 | 0.9361 | 0.3763 | 0.9361 |
| Application (1 January 2008 to 29 February 2008) | 60 | 1.0502 | 0.0146 | 5.0863 | 0.9025 | 0.4702 | 0.9012 |
| Lianshiwan | | | | | | | |
| Calibration (23 October 2007 to 31 December 2007) | 70 | 3.1534 | 0.2342 | 8.2496 | 0.8694 | 0.8761 | 0.8694 |
| Application (1 January 2008 to 29 February 2008) | 60 | 3.2403 | 0.1633 | 8.8179 | 0.8593 | 0.8946 | 0.8339 |
| Guangchang | | | | | | | |
| Calibration (1 January 2008 to 29 February 2008) | 70 | 5.0674 | 0.8933 | 9.2304 | 0.7235 | 1.3283 | 0.7235 |
| Application (1 January 2008 to 29 February 2008) | 60 | 4.2807 | 0.3429 | 9.2829 | 0.8359 | 1.1329 | 0.8294 |

## 5. Conclusions

This study developed a new statistical model that can accurately predict the salinity variations in estuaries. This model is based on an autoregressive model and incorporates the Gamma distribution function to correct the hysteresis effects of salinity. This study also added the dynamic corrections of short-term effects, based on the fixed corrections of long-term effects, to weaken the hysteresis effects. The model was tested against salinity data collected at different pumping stations of the PRE. The comparison showed satisfactory prediction according to the statistical parameters of $R^2$, *RMSE* and *NSE*.

The value of river discharge was increased and decreased to study its effects on salinity. Results showed that when the river discharge decreases by 50%, the increasesin $S_Q$ and $S_t$ are significantly higher than the decreasesin $S_Q$ and $S_t$ in the case of a 50% increase in river discharge. Salinity variation is more sensitive to smaller river discharge. In addition, the reliability of salinity prediction for the next 7days was examined using river discharge of three levels: small (1400 m$^3$/s), medium (2000 m$^3$/s) and large (9000 m$^3$/s). In all three cases, the model accurately predicted the salinity variation.

Overall, this study proposed a new model based on comprehensive memory effects of river discharge and salinity, providing an accurate method of predicting salinity for effective water resources management. However, hourly changes in salinity cannot be predicted, and there are some limitations.

**Author Contributions:** Conceptualization, R.Y.; methodology, J.K.; software, R.Y.; validation, J.K., C.S.; formal analysis, J.Z.; investigation, J.K., W.Z.; resources, J.K.; data curation, R.Y.; writing—original draft preparation, R.Y.; writing—review and editing, J.Z.; visualization, C.S.; supervision, W.Z.; project administration, J.K.; funding acquisition, R.Y.,J. K. All authors have read and agreed to the published version of the manuscript.

**Funding:** This research was funded by National Key R&D Program of China (2019YFC0409004), the Qing Lan Project of Jiangsu Province(2020), the Open Research Foundation of Key Laboratory of the Pearl River Estuarine Dynamics and Associated Process Regulation, Ministry of Water Resources [2018]KJ07, the Belt and Road Special Foundation of the State Key Laboratory of Hydrology-Water Resources and Hydraulic Engineering (2018491111) and the Open Research Foundation of Key Laboratory of Coastal Disasters and Defense of Ministry of Education (201706).

**Conflicts of Interest:** The authors declare no conflict of interest.

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
