# Peer review of "An Alternative Statistical Model for Predicting Salinity Variations in Estuaries"

_sustainability, doi:10.3390/su122410677_

Round 1

Reviewer 1 Report

The article presents an interesting statistical model for predicting salinity variations in estuaries applied to the Pearl River Estuary.

What needs to be improved are the titles of figures and tables that are in some cases the same or too similar although represent different results.

From figures' and tables' titles it should be clear what is presented in them. (tables 2,3 and 4 have the same title; figures 7,9,13 should have more details in titles).

In Figure 5 the numbers of equations (8),(9),(10) should be checked.

Author Response

The article presents an interesting statistical model for predicting salinity variations in estuaries applied to the Pearl River Estuary.

What needs to be improved are the titles of figures and tables that are in some cases the same or too similar although represent different results.

Thanks for your advice. We have changed some titles of figures in the paper such as adding time to the figure to make it clearer. More details can be find in the presented ms.

From figures' and tables' titles it should be clear what is presented in them. (tables 2,3 and 4 have the same title; figures 7,9,13 should have more details in titles).

Thanks for your advice. We have changed some titles of figures in the paper such as adding time to the figure to make it clearer. More details can be find in the presented ms.

In Figure 5 the numbers of equations (8),(9),(10) should be checked.

Thanks for your advice. We have changed equations (8),(9),(10) to equations(5), (7) and (8)in Fig.5.

Reviewer 2 Report

  1. Clearly define what the small, medium and large discharge represent. e.g., does the small discharge represent 7Q10, 1Q10 or something else?
  2. Page 2, not for sure what the area 9750 km2 represent, the catchment of river or delta, the area of delta or the area of interest in this study.
  3. Page 2, should the "Pear River networks" be the "Pear River delta"?
  4. Figure 1, clearly highlight the study area and all names mentioned in ms.
  5. Figure 5, should the legends be Eq. (5), (7) and (8)?
  6. Figure 8, it is hard to differentiate the lines between increase and decrease 50%. Use different line types and colors.

Author Response

  1. Clearly define what the small, medium and large discharge represent. e.g., does the small discharge represent 7Q10, 1Q10 or something else?

Thanks for your advice. In this paper, small, medium and large discharge represent 1400 m3/s,2000 m3/s,9000 m3/s, respectively.

  1. Page 2, not for sure what the area 9750 km2 represent, the catchment of river or delta, the area of delta or the area of interest in this study.

Thanks for your advice. 9750 km2 represents the area of interest in this study.

  1. Page 2, should the "Pear River networks" be the "Pear River delta"?

Thanks for your advice and changes have been made.

  1. Figure 1, clearly highlight the study area and all names mentioned in ms.

Thanks for your advice. We have clearly highlighted the study area and all names mentioned in ms.

  1. Figure 5, should the legends be Eq. (5), (7) and (8)?

Thanks for your advice. We have changed equations (8),(9),(10) to equations(5), (7) and (8)in Fig.5.

  1. Figure 8, it is hard to differentiate the lines between increase and decrease 50%. Use different line types and colors.

Thanks for your advice. To make the difference of the lines between increase and decrease 50%, we have change the line types.